# The Assessment of White Matter Integrity Alteration Pattern in Patients with Brain Tumor Utilizing Diffusion Tensor Imaging: A Systematic Review

**DOI:** 10.3390/cancers15133326

**Published:** 2023-06-24

**Authors:** Aiman Abdul Manan, Noorazrul Azmie Yahya, Nur Hartini Mohd Taib, Zamzuri Idris, Hanani Abdul Manan

**Affiliations:** 1Functional Image Processing Laboratory, Department of Radiology, Universiti Kebangsaan Malaysia Medical Centre, Cheras, Kuala Lumpur 56000, Malaysia; p97676@siswa.ukm.edu.my; 2Diagnostic Imaging and Radiotherapy Program, Faculty of Health Sciences, School of Diagnostic and Applied Health Sciences, Universiti Kebangsaan Malaysia, Kuala Lumpur 50300, Malaysia; azrulyahya@ukm.edu.my; 3Hospital Universiti Sains Malaysia, Kubang Kerian 16150, Malaysia; nhartini@usm.my (N.H.M.T.); zamzuri@usm.my (Z.I.); 4Department of Radiology, School of Medical Science, Health Campus, Universiti Sains Malaysia, Kubang Kerian 16150, Malaysia; 5Department of Neurosciences, School of Medical Sciences, Universiti Sains Malaysia, Kubang Kerian 16150, Malaysia; 6Department of Radiology and Intervency, Hospital Pakar Kanak-Kanak (Specialist Children Hospital), Universiti Kebangsaan Malaysia, Cheras, Kuala Lumpur 56000, Malaysia

**Keywords:** diffusion tensor imaging, brain tumor, white matter tract microstructure, diagnostic, prognostic

## Abstract

**Simple Summary:**

Diffusion tensor imaging is a neuroimaging tool for non-invasively visualizing the white matter tract of the brain. For more than 20 years, studies have been discussing the possible white matter tract alteration that could be caused by the growth of a tumor. There are four main white matter tract morphology alterations: displacement, edematous, infiltration, and disruption. This alteration is mainly based on the white matter tract’s orientation and position and the diffusion tensor imaging parameter measurements. White matter tract alteration studies showed that they were not mutually exclusive in certain characterizations, and different types and grades of tumors have mostly depicted displacement or infiltration.

**Abstract:**

Alteration in the surrounding brain tissue may occur in the presence of a brain tumor. The present study aims to assess the characteristics and criteria of the pattern of white matter tract microstructure integrity alteration in brain tumor patients. The Scopus, PubMed/Medline, and Web of Science electronic databases were searched for related articles based on the guidelines established by PRISMA. Twenty-five studies were selected on the morphological changes of white matter tract integrity based on the differential classification of white matter tract (WMT) patterns in brain tumor patients through diffusion tensor imaging (DTI). The characterization was based on two criteria: the visualization of the tract—its orientation and position—and the DTI parameters, which were the fractional anisotropy and apparent diffusion coefficient. Individual evaluations revealed no absolute, mutually exclusive type of tumor in relation to morphological WMT microstructure integrity changes. In most cases, different types and grades of tumors have shown displacement or infiltration. Characterizing morphological changes in the integrity of the white matter tract microstructures is vital in the diagnostic and prognostic evaluation of the tumor’s progression and could be a potential assessment for the early detection of possible neurological defects that may affect the patient, as well as aiding in surgery decision-making.

## 1. Introduction

The challenge of understanding how tumor cell growth influences the surrounding brain cells has always been the key in understanding the pathophysiology of brain tumor invasion. A neuroimaging technique, such as diffusion tensor imaging (DTI), is used to visualize the white matter tracts (WMT) in vivo. DTI will provide additional information on the WMT integrity and its microstructural architecture in the presence of the tumor [1,2,3]. A lot of research has been conducted in this area, focusing on various tumor structures and histopathologies. The finding suggested that the tumor invasion impacts the patient’s neurocognitive and motor function, coordination, speech, and ultimately affects the patient’s quality of life [3,4,5,6].

Integrating neuroimaging tumor analysis could be the alternative step for tumor diagnostics, eventually leading to faster information for cancer treatment. The visualization of the white matter tract through DTI could provide potentially important additional information for a successful diagnostic and the prognostic utility of the brain tumor patient, particularly in terms of a preoperative, surgical approach and potential neurological deficiency in the postoperative [4,6].

Many studies have been conducted to evaluate the alteration of white matter tract integrity patterns; however, the information was inconclusive. The invasion of the brain tumor affected the surrounding brain tissue, including the white matter tract, which altered the microstructures of the tract. The study on characterizations could be seen to have started nearly 20 years ago. Chronologically, the publication of the WMT characterization pattern using diffusion tensor imaging was first introduced by Witwer et al. in 2002 and has become the primary reference for subsequent studies [1]. Four major pattern characteristics are recognized in the white matter tract morphology and alteration of the microstructure integrity: displacement, edematous, infiltration, and disruption [1,2,7]. We have seen the utilization of WMT patterns in research on the different grades and types of tumors, and more precisely, on the high-grade gliomas (HGG) and low-grade gliomas (LGG) that were in abundance. The discussion on this matter has been dispersed. Furthermore, research on the different cancer pathological influences on WMT alterations were still lacking.

This study aims to systematically evaluate the white matter tract microstructural integrity characterization and the factors that contribute to its pattern under one roof, mainly on the type of brain tumor and the tumor grade, by utilizing DTI as the neuroimaging technique. To the best of the authors’ knowledge, this systematic review study is the only one to evaluate and discuss these different patterns—which are the four major microstructural integrity changes and the factors involved, primarily in the histopathology of the tumor—in depth. This review comprehensively explains the interaction between brain tumor growth and white matter tract microstructure integrity for previous, current, and future studies.

## 2. Methodology

### 2.1. Guidelines

This systematic review was conducted following the Preferred Reporting Items for Systematic Review and Meta-Analyses (PRISMA) guideline and was checked using the PRISMA checklist. These checklists are recorded in Appendix A. This method was followed according to the prior studies [6,8,9,10,11,12,13,14].

### 2.2. Search Chain

An extensive advanced search was conducted on 5 January 2023, by A.A.M., using three electronic databases: National Centre for Biotechnology Information (PubMed), Scopus, and Web of Science (WoS). The literature lists were recorded, screened, and analyzed from their titles, abstracts, and full articles. The main search was for studies focusing on the morphological changes of the white matter tract in the presence of brain tumors. The search was conducted using the keyword terms: “diffusion tensor imaging or diffusion tensor tractography or diffusion tensor or tractography or fiber tracking or fibre tracking”; “brain tumor or brain tumour or brain lesion or brain cancer or brain neoplasm”; “preoperative or pre-planning or presurgical or pre-surgery”; “white matter tract integrity or white matter tract morphology or white matter tract microstructure or white matter tract topology or white matter tract characterization or white matter tract characterisation or white matter criteria or white matter tract alteration or white matter tract changes”. The full details of the search strategy using the databases of all three websites are shown in Appendix A.

### 2.3. PICOS, Inclusion and Exclusion Criteria

The selection of the literature was based on the PICOS criteria, which is abbreviated for patient, intervention, comparison, outcome, and study design, and involves selective inclusion and exclusion criteria. This step was conducted to remove irrelevant articles and confirm the eligibility of the study. In short, the literature reviewed focuses on brain tumor patients (P) who had undergone diffusion tensor imaging preoperatively (I); studies that recorded their analysis based on factors that influence the changes of WMT integrity, i.e., tumor types, tumor grades, and the measurement of DTI indices (C); studies that reported the evaluation of the pattern of morphology changes in the involved white matter tract in relation to the tumor (O); and were clinically studied (S). These PICOS criteria are simplified in Table 1.

The selection was limited to adult human subjects, English-language publications, and online availability. There was no restriction on the publication year. Review papers, case reports, and research papers of studies on the preoperative stage of a brain tumor that did not specify the white matter tract characteristics of morphological changes, as well as studies that included the use of other imaging modalities, were excluded. The number of patients was limited to more than 5 persons, as this could distinguish the selected papers from the case reports, which excluded Wei et al. (2004) and Laundre et al. (2005) [15,16].

### 2.4. Eligibility Assessment

To ensure the robustness of the literature selection, the bibliographic references and citations of the included literature were extracted using Google Scholar and screened for any related articles. Two independent reviewers (H.A.M. and N.Y.) reviewed all the selected literature. All selected articles were checked for their quality using the Study Quality Assessment Tools by the National Heart, Lung and Blood Institute (NHLBI) website, https://www.nhlbi.nih.gov/health-topics/study-quality-assessment-tools (accessed on 7 February 2023), recorded in Appendix A. The data were extracted and tabulated in the tables presented in Section 3. This extracted information was reviewed and discussed.

## 3. Results

### 3.1. Article Selection

The electronic database advanced search based on the keywords produced 973 articles from Scopus, 921 articles from PubMed, and 881 articles from WoS, making up a total of 2775 articles. The remaining articles were 1501 after duplicates were removed. In the first phase, these remaining articles were reviewed based on their title and abstract, leaving a total of 325 articles. Then, in the second phase, full articles were screened through the selection process, and the number of selected articles was cut down to 151. The selected papers were assessed for eligibility based on the PICOS and exclusion and inclusion criteria, leaving a total of 22 papers. For reassurance, the final selected article reference lists and citations were also screened, and other sources and two related articles were obtained [17,18]. The PRISMA flowchart of 2020 [19] was drawn, detailing the study selection, shown in Figure 1. One paper was taken from a previous study [6,20]. Finally, a total of 25 published papers were selected. All of the selected articles were assessed as reasonably good and fair using the NHLBI study quality assessment tools. This systematic review was registered under the International Prospective Register of Systematic Reviews (PROSPERO), with ID number CRD42023395743.

### 3.2. Extraction and Tabulation of Data

The 25 selected papers were analyzed. The data were extracted to the main objective on the white matter tract alteration patterns based on the factors that could influence its changes, such as the different types of tumors, grading, and the locations of the tumor [1,7,17,18,20,21,22,23,24,25,26,27,28,29,30,31,32,33,34,35,36,37,38,39,40]. Table 2 summarizes the demographic data of the patients and participants recorded in these papers, and includes the information on tumor type, white matter tract involved, DTI scanner and parameter acquisition, and pre-processing data information, which included the fiber tracking software and the visualization method for the pattern analysis, as well as the type of analysis. The objective of the papers was included to better understand the utilization of DTI visualization patterns in brain tumor analysis. The descriptions of the DTI parameter and tract orientation for the characterization of the white matter tract pattern alterations are compressed in Table 3. The analysis of the morphology of white matter tract alterations in the type of individual tumor was collected in Table 4, and its assessment in different groups in Table 5. Finally, the summarization of the outcome utilization of the DTI WMT characterization pattern alteration in preoperative and clinical settings is shown in Table 6.

### 3.3. Demographic Data and Summarization of Selected Papers

These selected papers comprised retrospective [1,7,17,20,21,22,23,24,25,26,27,28,29,30,31,32,33,34,35,36] and prospective studies [18,36,37,38,39,40]. A total of 1163 patients ranging in age between 2 and 85 were studied in the 25 selected papers. In terms of analysis, only patients that was 17 years of age or older were being considered. The articles comprised publications from 2002 to 2022 and were tabulated and analyzed in a synchronized timeline. Almost all of the papers involved all types of brain neoplasms and gliomas; three papers studied only brainstem cancer [20,22,26].

Most of the research applied fiber tracking analysis using the region-of-interest (ROI), mainly focusing on either a specific type of white matter tract, corticospinal tract [20,40], optic radiation [35] or geniculocalcarine (GCT) [37], pyramidal tracts [35] and inferio-fronto occipital fasciculus (Camins et al., 2022), or tracts in the vicinity of the tumor [21,23,25,30,31,34,38]. Some studies used both, including all major tracts in the brain [32,36]. Meanwhile, Chen et al. (2007) and Kovanlikaya et al. (2011) analyzed the fiber tract using the volume-of-interest (VOI), where multiple seed points in the expected anatomical region were drawn in a rectangular shape and kept at the same coordinate in the preoperative and postoperative evaluation [32,36]. Most of the papers used the 1.5 Tesla magnetic strength magnetic resonance imaging (MRI) machine [1,7,20,21,22,24,25] and in six papers used the 3.0 Tesla [17,26,28,32,33,38] whereas Schneider et al. (2021) used both MRI strength fields [35]. The methods of DTI pattern visualization were conducted using intensity of fractional anisotropy (FA) directional color-coded maps [1,7,20,21,22,24]; some also included diffusion tensor tractography [17,25,26,28,32,33,38]. Figure 2 illustrates the mode of the visualization of WMT through directionally encoded color-coded (DEC) FA maps and tractography.

The utilization of DTI in the assessment and characterization of the WMT alteration patterns is mainly to understand the interaction between the effect of the tumor growth on WMT morphology [1,7,21,25] and to provide information in preoperative and postoperative surgical decision-making [17,23,26,28,29,30,32,33,36,38] and in the assessment and prediction of postoperative neurological deficits [17,26,28,33,38]. Another objective includes testing the methodology of tractography compared to DEC maps and in the visualizing of the WMT pattern [18,35,40].

**Table 2 cancers-15-03326-t002:** Demographic data of the patients, tumor, white matter tract involved, DTI acquisition, and summarization of the study.

No	Author (Year) [Ref]	Patients; (Male, Female)	Range Age, (Mean) Years Old	Tumor Type	DTI Acquisition and Processing Data		White Matter Tracts Involved	Objective of the Assessment of the WMT Microstructure
					Scanner, Vendor, Gradient (mT/M)	Fiber Tracking Software: Visualization Method/ Type of Analysis		
Retrospective Study								
1	Witwer et al. (2002) [1]	9 (5 M, 4 F)	20–66 (44 years)	Intrinsic brain tumor	1.5 T, GE Signa, 40 mT/m	DTI analysis software toolbox (Research systems); Color-coded DTI maps/ NR	AF, CST, ILF, brainstem, OR, CC	Assessment of WMT characterization in relation to cerebral neoplasms in preoperative mapping.
2	Field et al. (2004) [21]	13 (NR)	20–66 (43 years)	Intracranial neoplasm	1.5 T, GE CVi Signa, NR	IDL (Research systems); Directional color maps/ ROI	WMT fiber tracts in the vicinity of the tumor	Assessment of characterization of DTI pattern of the WMT in vicinity of tumor using DT eigenvector directional color maps.
3	Yu et al. (2005) [20]	16 (12 M, 4 F) Control group: 24 (17 M, 7 F)	20–72 (51.7 years) Control group: 25–68 (52.5 years)	Brainstem	1.5 T, Siemens Sonata, NR	Siemens workstation, Leonardo; Tractography/ ROI	PT, OR, CC	Assessment the role of DTT in preoperative mapping for the surgical approach and postoperative assessment.
4	Lazar et al. (2006) [7]	6 (NR)	2–61 (NR)	Brain lesion	1.5 T, GE Signa, 40 mT/m	Tensor deflection algorithm; Tractography /ROI	WMT adjacent to lesion	Assessment of WMT postoperative, comparison between preoperative and postoperative DTI criteria.
5	Chen et al. (2007) [22]	10 (5 M, 5 F)	16–73 (41.9 years)	Brain stem tumor	1.5 T, Siemens Sonata, NR	BrainLab iPlan 2.5; Tractography /VOI	CST, ML	Assessment of DTI and WMT visualization for preoperative surgical planning and postoperative assessment.
6	Yen et al. (2009) [23]	43 (19 M, 24 F)	NR	Brain lesion	NR	VOLUME-ONE; Directionally encoded color maps /ROI	Tract adjacent to tumor, CST, OR	Assessment of characterization of WMT by DTI for preoperative viability or respectability of the tumor adjacent to WMT.
7	Nievas et al. (2010) [24]	8 (4 M, 4 F);	16–73 (53.25 years)	Solid posterior fossa tumors	1.5 T, Siemens Erlangen, NR	3D software (Siemens); Tractography /ROI	Brainstem-assessed fiber, CST, traverse pontine fibers lateral and medial lemniscus.	Assessment of WMT alteration between preoperative and postoperative DTI criteria of PFT.
8	Itagiba et al. (2010) [25]	44 (24 M, 20 F)	3–88 (44 years)	Intracranial neoplasms	1.5 T, Avanto, Siemens; NR	DTI Task Card (Siemens); FA Color-coded map /NR	WMT vicinity to tumor	Assessment of different patterns of characterization of WMT in brain tumor pt. using DTI, and its differential diagnosis utilization.
9	Kovanlikaya et al. (2011) [26]	14 (7 F, 7 M);	5–65 (28.7 years)	Brain stem tumor	3.0 T, Philips Achieva; NR	PRIDE V4—Fiber tracking 6.1, Phillips; Tractography /VOI	CST	Assessment of DTI/DTT on CST alteration in brainstem tumor and its comparison on preoperative and postoperative.
10	Bagadia et al. (2011) [27]	50 (33 M, 17 F);	24–77 (48.5 years)	Intra-axial brain lesion	1.5 T, NR, NR	Ge Healthcare workstation; Tractography /ROI	CST, Optic pathways, arcuate fasciculus	Assessment of preoperative DTI for surgical planning and postoperative prognostic.
11	Castellano et al. (2012) [28]	73 (27 F, 46 M);	9–70 (44.2 years)	Glioma	3.0 T, Philips Intera, 80 mT/M.	DTI studio, version 2.4.10; Tractography /ROI	CST, IFOF, SLF	Assessment of DTI tractography in preoperative in predicting EOR in glioma resection.
12	Ibrahim et al. (2013) [29]	32 (24 M, 8 F);	1–74 (32.8 years)	Intracranial neoplasm	1.5 T, Gyroscan Intera, Philips; NR	Philips medical EWS, Pride; Tractography and Color coded FA maps; /ROI	WMT involved in vicinity of the tumor	Assessment DTI characterizing of WMT in relation to brain neoplasm and its utilization in preoperative.
13	Farshidfar et al. (2014) [30]	10 (7 M, 3 F);	36–60 (48.6 years)	Intra-axial brain tumor/Cerebral gliomas	1.5 T, Siemens, Espree; NR	DTI studio software Tractography /ROI	WMT tract vicinity to the tumor, CC, CST, SFC, UC	Assessment of DTI-FT in preoperative planning and in treatment strategy technique of brain tumor patient based on the WMT criteria in Iran
14	Deilami et al. (2015) [31]	12 (NR)	NR	Intracranial lesions	1.5 T, Siemens, 45 mT/m	MedINRIA (version 19.0); Color coded maps/ ROI	Peritumoral region	Defining diagnostic cut-off for differentiating four major types of peritumoral WM involvement using FA
15	Zhukov et al. (2016) [17]	29 (13 M, 16\W)	NR (45 years)	Supratentorial tumor	3.0 T, NR, NR	MRI machine software; Tractography /ROI	PT	Investigation on the relationship between different tumor’s histology types and WMT
16	Dubey et al. (2018) [32]	34 (21 M, 13 F);	17–70 (48.3 years)	Intra-axial brain tumor	3.0 T, Phillips Ingenia, Phillips; NR	DTI Studio; Tractography and DEC /ROI	CST and major subcortical tracts	Assessment preoperatively the integrity and location of WMT and plan surgical corridor
17	Yu et al. (2019) [33]	17 (9 M, 8 F);	29–72 (53.2 years)	Intracranial lesion	3.0 T, Ingenia, Phillips, NR	Philips Extended MR Workspace; Tractography /ROI	CST	Assessment of preoperative surgical planning, the intraoperative evaluations and clinical outcome prognosis based on DTI-FT
18	Leroy et al. (2020) [34]	11 (8 M, 3 F)	27–68 (43 years)	Glial tumors	1.5 T, General Electric; NR	Volume viewer 11.3 Ext 8, GE; DEC/NR	Peritumoral tracts	Assessment of the correlation between the preoperative DTI tractography and histology mainly in fiber directional and tumor related fiber destruction.
19	Schneider et al. (2021) [35]	25 (NR)	30–85 (57.1 years)	Space-occupying intracranial lesion.	1.5 T, or 3.0 T GE siemens; NR	FuncTool or FiberTrak; CCM and Tractography/ROI	PT, OR	Assessment of pattern WMT using color -coded maps versus tractography.
20	Bakhshi et al. (2021) [36]	77; (54 M, 23 F)	NR (40.7 ± 14.8 years)	Intra-axial brain tumor	1.5 T: NR;NR	NR Tractography /ROI	Whole brain, most white matter tracts	Assessment of pattern involved WMT morphological changes by intra-axial brain tumor
**Prospective study**								
21	Zhang et al. (2017) [37]	32 (17 M, 15 F) Control 30 (15 M, 15 F)	35–61 (44.1± 3.6 years) 20–63 (39.2 ± 3.3 years)	Occipital neoplasm	1.5 T, Signa Twin, GE; NR	Volume One 1.72 and diffusion Tensor Visualizer; DTT and DEC /ROI	GCT	Assessment of the disruption of GCT in different occipital neoplasm by DTI
22	Khan et al. (2019) [38]	128 (78 M, 50 F)	16–82 (49 years)	Intra-axial supratentorial brain tumor	3.0 T, Philips Ingenia; NR	DTI studio; Tractography and color-coded map /ROI	WMT and fascicles involved in vicinity of the tumor	Assessment preoperative DTI planning in term of surgical corridor and outcome, tumor characterization and postoperative prediction according to the DTI criteria
23	Shalan et al. (2021) [39]	20 (14 M, 6 F)	20–55 (NR)	Brain gliomas	1.5 T, GE sigma, NR	Offline workstation; Tractography and color-coded map/ROI	CST, SLF, IFOF, CC, UC	Assessment of utility of DTI tractography as image technique and neurosurgery brain gliomas planning
24	Wende et al. (2021) [40]	14 (6 M, 8 F)	30–70 (50.1 ± 4.0 years)	Intracerebral tumor	3.0 T, Ingenia, Phillips, NR	MRtrix3; Tractography/ ROI	CST	Assessment of reliable FA cut-off tractography of CST
25	Camins et al. (2022) [18]	34 (18 M, 16 F)	22–71 (48 years)	Temporal insular tumor	1.5 T, Phillips Intera or Achieva; NR	Phillips Intellispace portal vers 10; DEC and tractography/ROI	IFOF	Assessment of preoperative IFOF involvement predetermined predictable patterns

3D: 3-dimentional, AF: arcuate fasciculus, CC: corpus callosum, CCFA: color coded fractional anisotropy, CCM: color-coded maps, CST: cortical spinal tract, DEC: directional encoded color maps, DT: diffusion tensor, DTI-FT—diffusion tensor imaging fiber tracking. DTI: diffusion tensor imaging, DTT; diffusion tensor tractography, EOR: extent of resection, EWS: extended workstation, F: female, FA: fractional anisotropy, FACT: fiber assignment by continuous tracking algorithm, GBM: Glioblastoma, GCT: Geniculocalcarine tract, GQI: generalized Q-sampling, HGG: High grade gliomas, IDL: interpretable deep learning, IFOF: Inferio fronto-occipital fasciculus, ILF: inferior longitudinal fasciculus, LGG: Low grade gliomas, M: male, MD: mean diffusivity, MR: magnetic resonance, mT/M: Militesla per meter, ML: Medial Lemniscus, NR: not recorded, OR: optic radiation, PFT: solid posterior fossa tumors, PNET: primitive neuroectoderm tumor, PT: pyramidal tract, pt.: patients: region of interest, SFC: superior frontal connecting fibers, SLF: superior longitudinal fasciculus, T: Tesla, UC: uncinate fasciculus, VOI: volume of interest, WMT: white matter tracts.

### 3.4. The Characterization of White Matter Tract Integrity Visualized by DTI

These studies highlighted various criteria and characterizations of white matter tract integrity and structure morphology by evaluating the DTI and tractography data, yet several similarities were observed. Most criteria were divided into five categories: non-affected; displaced or deviated; edematous; infiltrated; and disrupted, as described in Table 3.

The main criteria for deciding the pattern of WMT characterization are the orientation of the tract in the visualization of white matter tracts based on DEC FA maps and the tractography, as well as the FA and apparent diffusion coefficient (ADC) values of the tracts and the surrounding brain tissue. In short, displaced is when a WMT maintains normal anisotropy corresponding to its healthy contralateral hemisphere, but is located with an abnormal orientation on the color-coded map; edematous is when it has maintained normal anisotropy and orientation, but demonstrated high signal intensity on T2-weighted MR images; infiltrated is when it has reduced anisotropy but remained identified on orientation maps; lastly, disrupted is when the anisotropy is reduced and the tract could not be identified on orientation maps [1,2,7]. For normal tracts or non-affected patterns, the tract was seemingly intact, located far from the tumor, and the fascicle pathway and tract thickness were unaltered [17,22].

On the other hand, Field et al. (2004) and Shalan et al. (2021) derived the criteria in relation to FA and ADC quantitative differences [21,39]. To summarize, their idea of fiber tract pattern 1 could be characterized by normal or mildly decreased FA and mildly increased ADC of less than 25%, while patterns 2 and 3 showed substantially decreased FA and increased ADC; however, pattern 3 tract visualization depicted that the abnormal hues were not attributed to the mass bulk. In all of these patterns, the relationship between the FA and ADC was significantly inverse. Pattern 4 was distinguished by the absence of discernible tracts and isotropic diffusions. These patterns could be the exact derivative of the patterns of displacement, edematous, infiltration, and disruption.

Bakhshi et al. (2021) evaluated that the classification of the FA was specified in the exact numerical measurement of the FA for the classification derived by Witwer et al. They classified the WMT characterization based on the degree of FA, which, when unaffected, was in the range of 0.163 to 0.286; when displaced, the FA was 0.085 to 0.093; when edematous, the FA was 0.092 to 0.149; infiltrated when the FA in the range of 0.05 to 0.059, finally, disrupted, the FA was less than 0.05 [36].

**Table 3 cancers-15-03326-t003:** Determination of characterization and criteria of WMT pattern based on DTI indices parameter, quantitative value and percentage, and visualization orientation for the WMT.

Pattern	Anatomical Description	Indices/Parameters Used			
		FA (Quantitative)	FA	ADC	Percentage Difference
Normal	Not affected, fiber in the correct anatomical location.	0.163–0.286	Normal	Normal	NR
Displacement (Deviated/Deformed) /Pattern 1	Maintained normal anisotropy relative to the corresponding tract in the contralateral hemisphere but were situated in an abnormal location or with an abnormal orientation on color coded orientation maps.	0.085–0.093	Normal or mildly decrease	Normal or mildly increase	Nearly (<25%), for both ADC and FA
Edematous /Pattern 2	Maintained normal anisotropy and orientation but demonstrated high signal intensity on T2 weighted MR images	0.092–0.149	Decreased	Increased	NR
Infiltration /Pattern 3	Reduced anisotropy but remained identified on orientation maps	0.050–0.059	Decreased	Increased	NR
Disruption (Destroyed/Interrupted) /Pattern 4	Anisotropy was markedly reduced such that tracts could not be identified on the orientation maps	<0.050	Unidentified Isotropic/near isotropic	Isotropic, near isotropic	NR

Abbreviations: ADC: Apparent diffusion coefficient, FA: Fractional anisotropy, NR: not recorded.

### 3.5. Morphological Assessment of White Matter Tract Integrity with DTI DEC FA Map and Tractography

#### 3.5.1. Evaluation of the Individual Patients

Table 4 and Table 5 summarize the morphological changes and the assessment of the white matter tract integrity in relation to a brain tumor in two different classifications of patients, both individually and by groups. Table 4 represents the individual patient assessment of the morphological changes in the white matter tract integrity using the DTI data, DEC maps, and tractography data in relation to the tumor type. The characterization of the changes in the WMT for each patient was classified by tumor histology and graded according to the World Health Organization (WHO) grading [41,42,43]. The classification of the combined WMT patterns, which is the combination of two or more types of classification, were also included.

The following were the major patterns for the respective brain tumors from Table 4. The total number of patients used for this analysis was 806. Oligodendroglioma was the most commonly initiated infiltrated pattern, followed by displacement and disruption. Astrocytoma mostly showed displacement and disruption patterns, with a minimal pattern of infiltration, but showed the combination of the “infiltration and disruption” and “displacement and disruption” patterns. Oligoastrocytomas displayed that the patterns in the WMT were mostly displaced and infiltrated and have the combination of “displacement and disruption”. Meningiomas showed that the most commonly found pattern was the displacement pattern, with 77%, and infiltration with 8%. However, a 15% combination pattern of “displacement and disruption” was discovered. Gangliomas have been shown to have all cases in displacement. Glioblastoma (GBM) has shown an abundance of infiltration patterns, followed by displacement, and disruption. The most frequent combinations for the GBM tumor were “displacement and infiltration” and “infiltration and disruption” at 9% and 5%, respectively. Metastasis tumors recorded a 31% displacement pattern, acted on by the WMT, 14% in the combination of “infiltration and disruption”, and 13% in a disrupted pattern. For LGG, more than half of the sample showed a WMT displacement pattern, followed by infiltration. The combination that took up to 10% of the LGG total pattern was “infiltration and disruption”. In the case of HGG, 25% had an infiltration pattern on the WMT, 23% had an “infiltration and disruption” combination, 19% had a displacement, and 9% had a disruption pattern. It has been recorded in this study that 6% of the HGG had more than two patterns, including displacement, infiltration, and destruction [25].

**Table 4 cancers-15-03326-t004:** (a): Finding based on characterization of white matter tracts integrity in individual brain tumor patients’ histology and tumor grade. (b): Summarization of finding based on characterization of white matter tracts integrity in individual brain tumor patients’ histology and tumor grade in major brain tumor for the selected articles in percentage.

(a)											
No	Author (Year)	Characterization Assessment of White Matter Tracts (n = Number of Patient)									
		Patient (n)	Normal/Non-Affected	Displaced/Deformed/Deviated		Edema	Infiltrated		Disrupted/Destroyed		Others/More That 2 Characteristics
1	Witwer et al. (2002) [1]	9	-	Oligodendroglioma II (3), Malignant Oligoastrocytoma III (1)		Metastasis (1)	Oligodendroglioma III (1) Oligodendroglioma II (1)		-		**Displaced and Disrupted:** Pilocytic astrocytoma I (1) **Edema and Disrupted:** Astrocytoma IV (1)
2	Yen et al. (2009) [23]	28	-	Meningioma (4), Metastasis (1), Pontine glioma arachnoid cyst (1), GBM IV (2), Acoustic neuroma (2), PNET (1)		Metastasis (2), Gliomatosis cerebri (1)	Meningioma (1), GBM (2), Trigeminal neuroma (1), Gliomatosis cerebri (2)		Metastasis (2), Pontine glioma arachnoid cyst (1), Pilocytic astrocytoma (1), GBM (1), Oligodendroglioma (1), Ependymoma (1), Gliosis (1)		-
3	Itagiba et al. (2010) [25]	44	-	LGG (14)		-	-		-		**Displaced, Infiltrated, and** **Disrupted** GBM (12) Anaplastic astrocytoma (8/9) **Displaced and Edema** Metastasis (9)
4	Farshidfar et al. (2014) [30]	9	-	Astrocytoma I (1), Astrocytoma II (1), Oligodendrioma I (2) Oligodendrioma II (1),		-	-		GBM IV (1)		**Displaced and Edema** Oligodendrioma II (1) **Infiltrated and Disrupted** Oligodendrioma II (1), Astrocytoma II (1)
5	Lazar et al. (2006) [7]	4	-	Pilocytic astrocytoma I (1), Ganglioglioma I (1),		-	-		-		**Deviated and Infiltrated** Astrocytoma III (1) Astrocytoma IV(1)
6	Chen et al. (2007) [22]	3	-	Astrocytoma II (1),		-	-		-		**Deviated and Disrupted** Astrocytoma II (1), **Interrupted and Infiltrated** Astrocytoma II (1)
7	Kovanlikaya et al. (2011) [26]	7	Hemangio blastoma (1)	Diffuse fibrillary astrocytoma II (1) Pilocytic astrocytoma I (1) Astrocytoma II (1) Mixed neuronal glial I (1) Metastasis (1)		-	-				**Displaced and Disrupted** Anaplastic astrocytoma III (1)
8	Nievas et al. (2015) [24]	6	-	-		-	-				**Deviated deformed, thinning** interrupted Meningioma (2), Metastasis (2) **Deviated, deformed, thinning** Neurinoma (1) Metastasis (2) **Deviated thinning** Neurioma (1)
9	Bagadia et al. (2011) [27]	44	-	Oligodendroglioma (10) Metastasis (3) GBM IV (7) Anaplastic astrocytoma (4) Diffuse fibrillary astrocytoma (1) Pleomorphic xanthoastrocytoma (1) Pilocytic astrocytoma (1)		GBM IV (1)	GBM IV (3) Other (1)		Oligodendroma (1) GBM IV (1)		**Displaced and Edema** GBM IV (1) Metastasis (1) **Displaced and infiltrated** GBM IV (6) Oligodendroglioma (1) **Infiltrated and destroyed** GBM IV(1)
10	Yu et al. (2005) [20]	20 (3 Tracts)	-	OR Astrocytoma II (1), OR Astrocytoma III (1), PT GBM IV (1), OR Metastasis (1)		-	-		PT Astrocytoma II (1), CC Oligodendroglioma II (1), CC GBM IV (2) PT Metastasis (1) CC Metastasis (1)		**Displaced and disrupted** PT Astrocytoma III (1) PT Oligoastrocytoma III (3), CC Oligoastrocytoma III (2), PT GBM IV (2), PT Metastasis (2)
11	Deilami et al. (2015) [31]	12, (100 ROI)	-	LGG (7), HGG (15)		HGG (9)	LGG (15) HGG (36)		LGG (1), HGG (14)		
12	Zhukov et al. (2016) [17]	29	Glioma II (5), Glioma III (3), Glioma IV (3) Metastases (1)	Glioma I (2), Glioma II (1), Glioma IV (3) Metastases (1)		-	Glioma II (2), Glioma III (1), Glioma IV (5) Metastases (2)		-		
13	Khan et al. (2019) [38]	128	-	HGG (9), LGG (36), Metastasis (3)		-	-		-		**Infiltrated and Disrupted:** HGG (57), LGG (12), Metastasis (11)
14	Dubey et al. (2018) [32]	34	-	HGG (5), LGG (9), Metastasis (4)		-	-		-		**Infiltrated and Disrupted** HGG (12), LGG (3), Metastasis (1)
15	Zhang et al. (2017) [37]	32	-	Meningioma (6), Metastases (10)		-	-		-		**Displaced and Infiltrated** Glioma II (2) **Infiltrated and Disrupted** Glioma III and IV (7) **Displaced and Disrupted** Metastases (7)
16	Leroy et al (2020) [34]	11	-	-		-	GBM IV (1) Anaplastic oligodendroglioma III (3) Astrocytoma II (2)		GBM IV (1)		**Infiltrated and Disrupted** Astrocytoma II (1) GBM IV (2) Anaplastic oligodendroglioma III (1)
17	Shalan et al. (2021) [39]	44	LGG (2)	HGG (10), LGG (5)		HGG (7), LGG (4)	HGG (11), LGG (1)		HGG (4)		-
18	Bakhshi et al. (2021) [36]	77	Astrocytoma I (1), Metastasis (1)	Astrocytoma I (1), Astrocytoma II (1), Astrocytoma III (2), GBM IV (7), Oligoastrocytoma II (1), Oligoastracytoma III (2), Oligodendroglioma II (2), Oligodendroglioma III (2), Others (4)		-	GBM IV (17), Metastasis (2), Oligoastrocytoma II (1), Oligoastracytoma III (1), Oligodendroglioma II (10), Oligodendroglioma III (9), Others (5)		Astrocytoma I (1), GBM IV (2), Oligoastrocytoma II (1), Oligodendroglioma II (1), Oligodendroglioma III (1), Metastases (1) Others (1)		-
19	Schneider et al. (2021) [35]	25 (both DEC and TG)	Metastasis (1) DEC Glioma II (1)	Benign (2) Metastasis (3) Glioma II (2) Glioma III (2) TG Glioma IV (3)		-	-		Metastases (3)		**Displaced and Edema** TG Glioma IV (1) **Displaced and Disrupted** Metastasis (2) Gliomas IV (7) **Displaced and infiltrated** Glioma III (1) DEC Glioma IV (1) **Infiltrated and Disrupted** Glioma III (1) Glioma IV (1) **Edema and Disrupted** TG Glioma III (1) **Edema and Infiltrated** DEC Glioma III (1)
20	Camins et al. (2022) [18]	34	Pleomorphic xanthoastrocytoma II (1)	Ganglioglioma I (1), Diffuse astrocytoma II (1), Anaplastic astrocytoma III (1), GBM IV (2)					Anaplastic astrocytoma III (2), Anaplastic oligodendroglioma III (1) GBM IV (4)		**Edema and Infiltrated** Diffuse astrocytoma II (3), Anaplastic astrocytoma III (6), Gliosarcoma IV (1), Anaplastic oligodendroglioma III (4), GBM IV (7)
**(b)**											
		**Type of Tumor**								**Tumor Grade**	
**Pattern**		**Oligodendroglioma**		**Astrocytoma**	**Oligo** **astrocytoma**	**GBM**	**Meningioma**	**Ganglioma**	**Metastasis**	**LGG**	**HGG**
Normal									3.5% (3/86)	6% (10/180)	2% (6/353)
Displacement		36% (20/55)		54% (22/41)	36% (4/11)	29% (19/65)	77% (10/13)	(2/2) 100%	31% (27/86)	56% (101/180)	19% (66/353)
Edematous						2% (1/65)			3.5% (3/86)	2% (4/180)	5% (17/353)
Infiltration		43% (24/55)		5% (2/41)	18% (2/11)	35% (23/65)	8% (1/13)		7% (6/86)	18% (32/180)	25% (88/353)
Disruption		13% (7/55)		10% (4/41)	9% (1/11)	15% (10/65)			13% (11/86)	3% (6/180)	9% (32/353)
Non-exclusive											
Displacement + edema		2% (1/55)				2% (1/65)			12% (10/86)	<1% (1/180)	<1% (1/353)
Displacement + infiltration		2% (1/55)		7% (3/41)		9% (6/65)			1% (1/86)	2% (3/180)	1% (4/353)
Displacement + disruption				10% (4/41)	36% (4/11)	3% (2/65)	15% (2/13)		13% (11/86)	1% (2/180)	5% (16/353)
Infiltration + edema										2% (3/180)	5% (19/353)
Infiltration + disruption		2% (1/55)		12% (5/41)		5% (3/65)			14% (12/86)	10% (18/180)	23% (81/353)
Disruption + edema		2% (1/55)		2% (1/41)							<1% (2/353)
Others									2% (2/86)		6% (21/353)

Abbreviations: CC: Corpus callosum, CST: cortical spinal tract, DEC; Directionally color coded map, DTT; diffusion tensor tractography, GBM: Glioblastoma, HGG: High grade gliomas, LGG: Low grade gliomas, OR: Optic radiation, PNET: primitive neuroectoderm tumor, PT: Pyramidal tract, ROI; Region-of -interest, TG; Tractography. (a) These patients’ information’s were gathered from the selected articles and patients will be excluded based on the exclusion criteria i.e.,: ages below 18 and diagnose with other than brain neoplasm. Itagiba et al., Yu et al., Deilami et al., Schneider et al. patients selected were determined based on specific criteria. The papers published by Yen et al. (2009), Nievas et al. (2015), and Bagadia et al. (2011) were excluded from the recorded tumor grade due to insufficient data for each individual patient’s tumors grading. (b) The type of tumors have included all tumor grade. Percentage %: (number of patients involved/total patients)/100.

#### 3.5.2. Evaluation of the Patients in Groups

Table 5 summarizes the assessment of the morphological changes in the WMT integrity based on the pattern characterization reported in the group. Here, the classifications of tumor type and tumor grade, tumor location, and the diffusion tensor imaging parameter were specifically studied. In most of the papers studying patterns related to tumor grading, it was found that LGG or benign tumors have a higher association with the displacement pattern compared to HGG or malignant tumors, which were more closely associated with disruption and infiltration [29,32,34,36,38,39]. In addition, Leroy et al. (2020) reported that metastases also have more association patterns with displacement [34]. The only tumor type discussed was by Bakhshi et al. (2021), and they found that oligodendrogliomas often infiltrated the white matter tract. It has been argued that the involvement of the WMT in oligodendrogliomas was not associated with the tumor grade [36].

The presence of a brain tumor in a specific location influences the pattern displayed by the white matter tracts. In the tumors that were in the vicinity of the cortex, presumably in the range of less than 5 mm, the tract has been said to be partially interrupted. Tracts in close proximity to the tumor, in the range of 2 to 21 mm, were found to be displaced [33]. A different type of white matter tract feature would result in a different WMT morphology alteration. According to Yu et al. (2005), the pyramidal tract has a longer and wider range of fiber, which causes the tract to be more displaced and disrupted. In contrast, the corpus callosum, which has a short and small range of tract fiber type, was found to display a disruption pattern [20]. The tumor location also influences the displacement pattern, or more specifically, the direction of the displacement of the tracts. Displacement patterns have been shown to depend on the main tumor location in the brain, mainly whether the tumors were in the temporal lobe or the involvement of the insular cortex, as reported by Camins et al. (2022). According to their study, the tumor showed a displacement pattern, but the lateral tumor tended to displace medially, the medial tumor to the lateral direction, and the insular tumor tended to have medial displacement [18].

As described in Table 2, some of the papers discussed the cut-off point relative to the FA calculations, which is one of the parameters of DTI in the pattern of WMT alteration, which could be the indicator for classifying the pattern. Deilami et al. (2015) and Yen et al. (2009) studied the percentage of FA changes between the affected tract and the contralateral, unaffected hemisphere [23,31]. Yen et al. (2009) showed the sole characterization of each WMT that was being assessed. They found that in the calculation of the percentage of FA decrement compared to the normal hemisphere, only the disruption pattern showed a significant difference. In addition, edematous and disruption caused an FA decrement significantly less than displacement [23]. In a recent study, Deilami et al. (2015) discovered that the percentage decrement of the FA was more than −35% for displaced and edematous fiber and less than −35% for the majority of disrupted fibers. This is in agreement with Yen et al. (2009), who showed a 30% decrement; however, the paper by Yen et al. (2009) included infiltration into the 0% to −30% range. Infiltration fiber was scattered in the distribution of the cut-off by Deilami et al. (2015), and similar to Wende et al. (2021), infiltration fiber bundles, in terms of the FA cut-off value, should be vigilant, as 0.1 should also be considered, and it required a lower FA cut-off value [23,31,40]. Deilami et al. (2015) concluded that disruption was the most presumptive cut-off point [31]. The investigation of the utilization of DTI in determining the WMT characterization pattern of its microstructure integrity in preoperative and clinical settings is summarized in Table 6.

**Table 5 cancers-15-03326-t005:** Findings based on the characterization of white matter tract integrity based on the grouped patients.

Author (Year) [Ref]	Assessment of White Matter Tracts Characterizations and Microstructure Integrity
	Tumor Type and Grade
Ibrahim et al. (2018) [29]	Prevalence of disruption was higher in malignant compared to benign tumor group. Prevalence of displacement was higher in benign tumors compared to malignant tumors.
Deilami et al. (2019) [31]	Infiltration was the major pattern for HGG and LGG. Edematous comprised the minority.
Khan et al. (2019) [38]	Sig. correlations of HGG associated with disruption or infiltration, as well as metastasis. LGG mainly with displaced fibers
Dubey et al. (2019) [32]	HGG have showed to be more infiltrated or disrupted, meanwhile the LGG and metastasis have more association with displacement.
Leroy et al. (2020) [32]	Higher grade glioblastoma has higher proportional of destruction WMT
Shalan et al. (2021) [39]	HGG showed higher percentage of infiltration and disruption pattern than LGG
Schneider et al. (2021) [35]	Difference of pattern visualize to be compared between DEC and tractography was showed in the 6 of the studies, one case showed no useful in tractography. There is no correlation between pathology of the tumor in the method of visualization of WMT pattern using DEC and TG.
Bakhshi et al. (2021) [36]	High grade astrocytoma cause infiltration of WMT, low grade astrocytoma caused displacement, and oligodendroglioma tumor type often infiltrated WMT. The involvement of WMT in oligodendroglioma was not associate with the grade tumor.
	**Tumor location**
Yu et al. (2005) [20]	Location of the tumor in vicinity to different type of WMT. Nature of pyramidal tract having longer fiber and larger range, resulting in more displayed of displacement and disruption. Corpus callosum have short fiber and small range of movement, resulting displayed disruption pattern
Yu et al. (2019) [33]	Partial interruption was evidence in lesion close to cortex, in the range of 0–5 mm. WMT closest to the proximity to the tumor was found displaced in all patients in the range of 2–21 mm.
Camins et al. (2022) [18]	Displacement patterns depended on main tumor location in temporal lobe, and insular involvement. All tumor showed superior displacement pattern. Lateral tumors tend to displace medially, medial tumor to laterally, and insular tumor tend to have medial displacement.
	**DTI parameter (cut-off points)**
Yen et al. (2009) [23]	Significant ΔFA (FA changes) between affected hemisphere and control contralateral WMT only in disruption Edematous and disruption ΔFA are sig. less than displacement ΔFA Positive percentage of ΔFA was associated with edematous and displacement, 0% to −30%, is likely associated with displacement or infiltration, and the percentage less that −30%, was associated with WMT disruption
Deilami et al. (2015) [31]	ΔFA% was more than −35 for displaced and edematous fiber and less than −35 for the majority of disrupted, but infiltration fibers scattered distribution. Mean ΔFA was the least for disruption, infiltration, edematous, and displaced parts. Disruption the most several presumptive cut-off points.
Wende et al. (2021) [40]	FA cut off in infiltrated fiber bundles, a difference of 0.1 value should be considered. Infiltrate CST trigger vigilance and may require lower cut-offs.

CST: cortical spinal tract, DEC: directional color-coded map, DTI: Diffusion tensor imaging, FA: fractional anisotropy, HGG: High grade gliomas, LGG: Low grade gliomas, mm: millimeter, Pt: patients, Sig: significant, TG: Tractography, WMT: white matter tract.

**Table 6 cancers-15-03326-t006:** Summarization of the studies outcome to the WMT morphology alteration.

Author [Ref]	Summarization of the Outcome to the WMT Tract Morphology Alteration
Yu et al. [20]	Surgical approach different based on the pattern of WMT. Maximizing the resection in simple displacement, to enlarge the extent of tumor resection while preservation of displaced part in displaced with disruption and maximized tumor while preserved the residual part in disrupted tracts.
Bagadia et al. [27]	Pt. with pure displacement had the best postoperative prediction outcome, while those with infiltration had a poorer outcome
Castellano et al. [28]	In preoperative tumor volume less than 100 cm^3^, intact fascicles higher probability of total resection. In preoperative tumor volume more than 100 cm^3^, infiltration or displaced fascicles predicted partial resection or subtotal.
Dubey et al. [32]	Total resection achieved in 61.2% of displacement pattern tract and 31.2 % in infiltration or disruption pattern of the tract. DTI have given crucial information of the infiltration and displacement course.
Khan et al. [38]	Respectability of maximum safe resection was higher in pt. with displaced fibers and lower in those with disrupted/infiltrated fibers, statistically sig. Fewer pt. had neurologic deterioration in displaced, compared to disruption or infiltration
Leroy et al. [34]	LGG have higher preservation of subcortical fiber tracts. DTI sensitivity and specificity to predict disrupted fiber tracts were, respectively, of 89% and 90%.
Schneider et al. [35]	Postoperative improvement of the pattern was seen mainly in the disruption DEC showed more improvement of WMT to be compared with tractography, postoperatively
Bakshi et al. [36]	Infiltration tumors have poor functional outcome, low KPS score
Shalan et al. [39]	Postoperative evaluation of WMT pattern were improved in displaced and edematous meanwhile not the case for infiltration and disruptive pattern HGG and pattern disrupted have the higher subtotal resection
Wende et al. [40]	FA cut-off value of 0.15 for tractography for neurosurgery and shorten the TG workflow.
Camins et al. [18]	IFOF displacement pattern are reproducible

CST: cortical spinal tract, DEC: directionally color coded maps, diff: difference, DTI: diffusion tensor imaging, FA: fractional anisotropy, HGG: High grade gliomas, IFOF: inferior fronto-occipital fasciculus tracts, KPS: Karnofsky Performance score, LGG: Low grade gliomas, Pt.: patients, Sig: significant, TG: tractography, WMT: white matter tract.

### 3.6. Preoperative and Clinical Outcome of the WMT Characterization Assessment

Similar findings could be seen in most of the studies; namely, that knowing the characterization of the WMT morphology can assist the decision to extend tumor resection in neurosurgery. Total resections were higher in intact fascicles, including displaced tracts, but the probability of total resection for infiltration and disrupted tracts was low [20,28,29,32,38]. The postoperative assessment of the WMT microstructure showed an improvement compared to preoperative WMT assessment—mainly on displacement and edematous and compared to infiltration and disruption—making pure displacement the best postoperative prediction outcome, and infiltration and disruption the poorest [27,36,39]. In their paper, Camins et al. (2022) discussed the reliability of DTI as a prediction for understanding the relation of morphological WMT microstructure integrity. They concluded that the displacement of the inferior longitudinal fasciculus tract was predictable based on the location of the tumor, ensuring the utilization of the pattern in the DTI WMT morphology study [18]. In the study to assess the FA cut-off value in tractography, Wende et al. (2021) reported that a FA value of 0.15 was a reasonable FA cut-off and would have shortened the tractography workflow in the preoperative phase [40].

## 4. Discussion

The study of the differences in the characterization of the white matter tract microstructure is an essential step toward understanding the coexistence of the effect of tumor growth and the changes it causes in the white matter tracts and surrounding brain tissues. The visualization of brain tumor gliomas is known to be heterogeneous, resulting in a wide range of imaging qualities using MRI [44]. DTI, on the other hand, is a non-invasive tool for the in vivo visualization of white matter tracts. It is also used for the diagnostic evaluation of gliomas and for estimating the preoperative degree of decision-making on surgical resection. It is clear that the DTI pattern evaluation for preoperative planning has provided information on the possibility of the amount of tumor resection and has contributed to underlying the surgical corridor for a successful resection [18,28,32,33,34,38,39]. It is also important for predicting the neuro-deficit assessment of whether the relationship of different morphological changes in the white matter tract leads to neuro-deficit improvements in postoperative brain tumor patients [27,36,38,39].

The prognostic nature of DTI is primarily based on the correct estimation of total resection and the outcome of neurological deficits. Knowing the tract was intact but displaced allowed the surgeon to adapt the approach to preserve the tract during resection [21,28]. The intensity of tumor growth on the tract morphology changes could be a sign of extensive resection. Displacement could result in total resection. In line with the studies by Castellano et al. (2012) and Khan et al. (2019), small tumors with displaced fascicles showed good resectability. However, in the case of infiltration or disruption, the tumor would be partially resected [28,38]. It has been reported that HGG and metastases have a more significant impact on DTI tractography, which can cause simple disruption and worsen clinical outcomes [33,35]. Displacement tracts are reproducible and very reliable for surgery prognostic, as multiple studies have shown their prediction in the movement orientation of tracts [18,45].

The difference in word choices for certain criteria and different approaches to the criteria could be seen throughout the early publications. A closer look into the literature revealed that some suggested different classifications of white matter tract morphology based on their understanding and the objectives of their research. Nievas et al. (2010) proposed that the characterization of the WMT microstructure is evaluated by their spatial orientations; deviation or deformation; and integrity, whether there were thinning or interrupted [24]. Yu et al. (2005) showed three simple types of classification: type 1, simple displacement; type 2, displacement with disruption; and type 3, simple disruption, as similarly applied by Yu et al. (2019) [20,33].

Similarly, Lazar et al. (2006) indicated the different approaches to characterizing the pattern using tractography, in classifying the names deviated and deformed, which could be similar to displacement, and interrupted to disrupted. Lazar et al. (2006) indicated the same criteria; however, the author disputed the characterization of edematous, as agreed by several other authors [7,22,26,46]. Lazar et al. (2006) pointed out that these patterns were used in their studies mainly for brainstem tumors, as the benchmark would be the control participant with known anatomy [7]. Some included the control participants as an indication of comparison in their studies [20,37]. In brainstem cancer investigations, the contralateral unaffected hemisphere control was not used as an indicator for the differentiation of the contrast pattern. This is because the brainstem is very much non-bilateral [22].

The methodology of visualization has been evaluated, and it is an interesting side to look into. Previous research revealed that, in some cases, WMT microstructural integrity was evaluated using FA color-coded directional maps and tractography. The variation in dimensional visualization may aid in better describing the pattern: 2-dimensional eigenvector maps, which give a diffused directional movement based on a color-coded map; and as for 3-dimensional tractogram visualization, they have the ability to visualize the orientation of the overall tracts and can give the radiologist a clear depiction of the white matter tracts. Schneider et al. (2021) explained that the evaluation of axon bundles from DTI directional color-coded maps provided more assistance compared to tractography. However, these two methods have shown themselves to be equally good alternatives [35].

The characterization and classification of patterns introduced for the assessment of white matter tract integrity were based on the changes in the anisotropy or FA value and the orientation abnormalities that were being compared with the unaffected side [1]. Diffusion-weighted imaging detected the diffusion of free water molecules within the constricted spaces, i.e., the myelin sheath of the axon, creating anisotropic motions. This resulted in the calculation of the FA, which is the measurement of the normalized fraction of the tensor degree of anisotropy on a scale of 0 (isotropic) to 1 (anisotropic), and the ADC, which is the average of the three eigenvectors [4,36]. In the presence of a brain tumor, the FA is decreased, and the ADC is increased, which are inversely correlated, resulting from the destruction of the axonal integrity by tumor growth [21]. The FA range and FA maximum value can distinguish LGG from HGG, as LGG have a significantly lower value [47]. Distinguishing metastases from HGG is difficult on diffusion-weighted imaging; however, it was found that there is an increased FA peritumorally, and a significant decrease in the mean diffusivity (MD) in HGG compared to brain metastases, while intratumorally, no significant changes were seen [48,49,50]. Similarly, meningiomas showed a significantly higher FA and lower ADC compared to HGG [51]. In a study on the disruption of GCT in different occipital neoplasms, Zhang et al. (2017) found that the FA value showed no significant difference across tumor types; however, the MD value of the GCT in the metastasis tumor group was higher than the other groups [37]. An increment in the MD or ADC would be an indicator of an increase in the extracellular spaces and decrease in the tract cellularity [52]. This shows the important parameters to be looked at for differentiation; that is, the fractional anisotropy and mean diffusivity or apparent diffusion coefficient.

Despite decades of research, the issue of the exact characterization of white mater tract integrity and its effect on brain tumor growth is continuously debated among researchers. Some studies have called into question the properties of the distinction between the patterns of infiltration and edematous infiltration. Vasogenic edema surrounding the tumor could also infiltrate the tracts without changing the orientation [21,35], causing the widening of fiber bundles without disrupting the cellular membrane [2,22]. The researcher disregarded the need for distinguishing edema and infiltrated tissue because the criteria was not DTI-based. Other methods required distinctive edema, which is appropriate for clinical consideration, namely in MR T2-hyperintense or FLAIR imaging [7,23]. Recently, it has been suggested by an experienced neuroradiologist that the comparison could be conducted with visualization [18,34]. In other words, infiltration by lesion could not be differentiated from infiltration by edematous as it also causes a reduction in the FA value; only differentiation through tract orientation visualization is possible as the infiltration shows abnormal hues and edematous shows a normal location and direction, especially in displaced tracts [18,35].

Our finding on the individual patients pertaining to the main major pattern of WMT characterization for astrocytoma, meningioma, oligoastrocytoma, ganglioma, and LGG was displacement, and for oligodendroglioma, metastases, GBM, and HGG, this was infiltration. Celtikci et al. (2018) reported qualitative analysis through generalized q-sampling imaging (GQI) and also found that LGGs might be displaced and infiltrated, similar to this study’s finding [53]. For the tumor type analysis, there was no association to tumor grade, which means all grades were included. This was backed by the study by Bakhshi et al. (2021), which reported that the effects of oligodendroglioma on the WMT were not linked to any tumor grading, and mostly caused infiltrations, as it is diffusely infiltrating [36,54]. However, high-grade astrocytomas were said to be mostly infiltrated [36]. This finding could largely indicate that most high-grade gliomas were astrocytomas, and GBM was also known to be astrocytoma grade IV [55]. Gangliomas have been shown to be displaced, but the sample size was insufficient. Deilami et al. (2015) indicated that the major type of involvement for both LGG and HGG is infiltration, while edematous was the least [31]. LGG growth was slow, and often spared neural functions as they infiltrated [50]. In contrast, LGG was mostly found to be displaced. HGG, or glioma grade IV, was reported by most studies to mostly have the disruption pattern [18,28,32,34]. Metastasis is mainly associated with edematous pattern visualization as the theory of vasogenic edema was mainly found around metastases tumors [25].

The characterization of WMTs based on these criteria results in a non-exclusive pattern is comprehensively accepted by most of the studies reported on. The WMT characterization was not mutually exclusive, and as a result, two or more pattern combinations could be seen in a patient. This is because no simple correspondence between the pathology and imaging pattern have been found [21]. For example, meningioma and neuroma are defined as being non-infiltrative. However, this is likely to be incorrect because Yen et al. (2009) indicated that meningioma has been found to have infiltration on white matter tracts. The findings on individual tumor patients’ assessments showed that the characterization of the white matter tracts were not specific to one type of tumor [23]. Although some of the histological diagnoses of the tumors were complemented with certain DTI patterns, i.e., mostly edema observed in metastases and HGG showed infiltration patterns, the observation of combination patterns may have called into question the use of DTI in differential diagnosis [25].

The FA value cut-off point could be an indicator that determines the WMT pattern and characterization, where in this case, the percentage of changes in the FA were calculated. Different studies have come upon various cut-off values to distinguish between two pathologies [50]. The pattern that had several cut-off points was disruption, and it is considered the most aggressive process as the destruction pattern has more isotropic regions [31]. However, the DTI sensitivity and specificity in the prediction of disrupted tracts were in the range of 89% to 90% [34,40]. The FA cut-off value could be an indicator for WMT pattern characterization, as most of the positive FA change values could indicate displacement or edematous and would rule out the possibility diagnosis of disruption [23,31,40]. It has been reported by Wende et al. (2021) that a 0.1 value change could distinguish infiltration from displacement. A reliable FA cut-off value would be critical as a FA cut-off that is too high would result in annihilating the true anatomical part of the fiber bundle, which could differentiate between false positive and false negative results [40].

### Limitation

One limitation of this review is that it only evaluated the depiction of qualitative data in the visualization of the DTI tractography and the FA and ADC measurements, ignoring the other potential quantitative parameters of DTI. Other possible parameters to look into are the radial diffusivity and axial diffusivity [52,56]. The selected publications were in English, and we discarded those that were not. The selected studies only included those that characterized the patterns of WMT alteration; thus, we excluded the papers that did not specify these four patterns. Moreover, the publications that included combinations of other neuroimaging modalities were also discarded. The combination of DTI and functional MRI could be used to seed the ROI in areas of the brain that are displaced or edematous [57]. The use of diffusion perfusion imaging, diffusion kurtosis imaging (DKI), and generalized Q-ball imaging (QBI), as well as the combination technique of MR spectroscopy, is said to be better in evaluating edema and visualizing white matter tract misorientation [44,50,58,59]. It was reported that qualitative anisotropy through GQI could assist in differentiating infiltration from displacement [53]. We understand that, in terms of tumor histology, many factors have been overlooked, as these factors or parameters should have been studied. Understanding each tumor type’s histopathology and pathology in depth could be the next crucial objective for future research. The source of artefacts could also be an issue. This includes the mass effect of the tumor, significantly deteriorating factors, image distortion, and the elimination of susceptibility artefacts related to the calcium and/or blood product within the lesion [7,22,25,26]. In addition, flaws in the analysis of the DTI data could have occurred because DTI is user-dependent and bias cannot be eliminated [26].

## 5. Conclusions

This systematic review was carried out by investigating the relationship between tumors and the morphological changes of white matter tract microstructure integrity and its alteration, as influenced by various types of tumors and tumor grades. The visualization of DTI data using color-coded directional maps and tractography could be used to characterize white matter tracts. Although there was differentiation when declaring the morphological change criteria, the main evaluation would always be conducted by measuring the fractional anisotropy and evaluating the orientation of the tract by comparing it with the unaffected contralateral hemisphere. Non-mutually exclusive patterns of WMT characterization, which have been observed between different brain tumors and tumor grades, waived the opinion on a certain absolute pattern to certain tumor types or grades and could be determined with certainty. However, it was agreed that DTI could be the initial diagnostic step in understanding the different histopathologies of tumors that acted on the morphological changes of WMTs. The determination of the characterization of white matter tracts affected by brain tumor growth would most likely be a successful tool for understanding the sustainability of the microstructure and architecture of white matter tracts, particularly in preoperative neurosurgery and postoperative neurological function preservation.

## Figures and Tables

**Figure 1 cancers-15-03326-f001:**
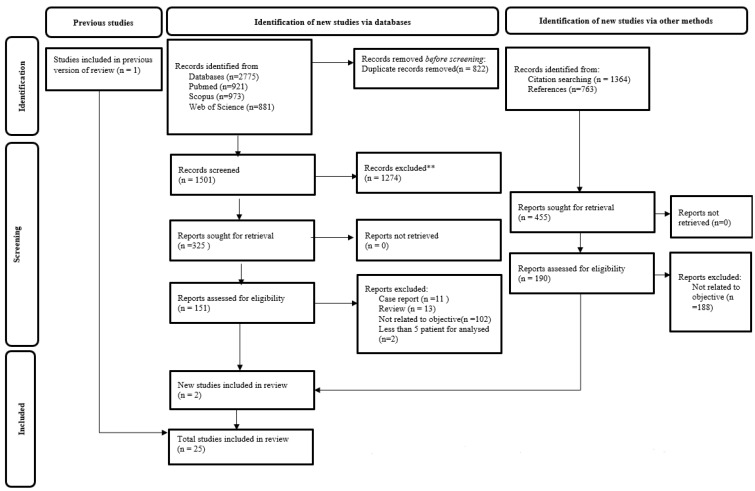
Flow chart of PRISMA 2020 on the process of the article selection. ** All record were excluded by human screening, based on the inclusion and exclusion criteria of the title and abstracts.

**Figure 2 cancers-15-03326-f002:**
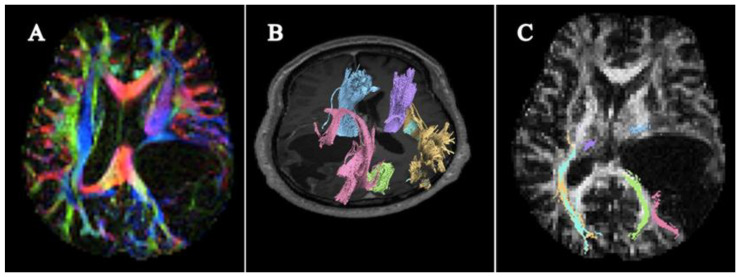
Mode of visualization of White matter tracts using Diffusion Tensor Imaging. Figure 2 illustrates the mode of visualizing WMT characterization using directionally encoded color-coded maps and tractography. (**A**) The axial view of the fractional anisotropy 2-dimensional directional color-coded maps of a brain tumor patient diagnosed with glioblastoma; (**B**) An axial view of fractional anisotropy maps obtained with 3-D tractography fibers; (**C**) Example tractography reconstruction of selected fibers. The tracts were colored in the following order: Blue and purple: corticospinal tract; light blue and green: optic radiation; yellow and pink: inferior longitudinal fasciculus. These figures were taken and reconstructed using DSI Studio using our bank data.

**Table 1 cancers-15-03326-t001:** The PICOS criteria of the systematic review.

PICOS	Criteria
Patient/Population	Adult Brain tumor
Intervention	Undergone diffusion tensor imaging
Comparison	Different Factors: Tumor types, tumor grades, DTI indices and parameter measurement
Outcome	Characterization on the morphological changes in the pattern of the involved white matter tracts integrity in relation of the tumor
Study Design	Original clinical published article

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
