# Peer review of "The Assessment of White Matter Integrity Alteration Pattern in Patients with Brain Tumor Utilizing Diffusion Tensor Imaging: A Systematic Review"

_cancers, 2023, doi:10.3390/cancers15133326_

Round 1
Reviewer 1 Report
It can be seen that in order to prepare this article, you have searched a large number of literature and done sufficient preparation work, which is worthy of recognition. But I have two suggestions for reference only. The first part is about the title of the article. From the current title, the author wants to evaluate the existing pattern of white matter integrity changes in patients with brain tumor. However, in the following, the author mainly introduces a series of studies based on DTI technology; But we know that there are currently multiple post-processing methods for the obtained DWI images, so can we make a modification to the title? This article is limited to a series of research reviews based on DTI, and other processing models will not be discussed temporarily. The second suggestion is about the table layout displayed by the author in the article. It is recommended that the author try to adjust the font size or other methods as much as possible to ensure that a word can be represented on one line. As shown in Table 4b, the letter "t" in the word "Displacement" occupies a separate row, making the layout of the table less perfect.
The authors have strong English language writing skills, and can clearly express their own meaning, while readers can also understand it well.
Reviewer 2 Report
Alteration in the surrounding brain tissue may occur in the presence of brain tumor.
The authors aim to assess the characteristics and criteria of the pattern of white matter tract microstructure integrity alteration in brain tumor patients.
They searched on Scopus, PubMed/Medline, and Web of Science electronic databases for related articles based on the guidelines established by PRISMA.
They detected Twenty-five studies on the morphological changes of white matter tract integrity based on the differential classification of white matter tract (WMT) patterns in brain tumor patients by diffusion tensor imaging (DTI).
The characterization was based on two criteria: the visualization of the tract, its orientation and position, and the DTI parameter, which were fractional anisotropy and apparent diffusion coefficient. Individual evaluations revealed no absolute, mutually exclusive type of tumor in relation to morphological WMT microstructure integrity changes. Different types and grades of tumors have mostly shown displacement or infiltration in most cases.
The authors concluded that characterizing morphological changes in the integrity of the white matter tract microstructures is vital in the diagnostic and prognostic evaluation of the tumor’s progression and be a potential assessment for early detection of possible neurological defects that may affect the patient, as well as aiding in surgery decision-making.
This is an interesting and well written systematic review.
I have only some minor suggestions:
1. Aim “Our objective is to systematically evaluate the white matter tract microstructural integrity characterization and the factors that contributed to its pattern under one roof. …” I suggest a more effective purpose. Use bullet points with sub aims if you need
2. The methods and results are arranged into paragraphs. I suggest to introduce the pars using some text before.
3. Avoid the use of we and our
Reviewer 3 Report
The authors provide a comprehensive and well written and clearly structured systematic review on white matter integrity alteration pattern in brain tumors.
The systematic review was conducted following the PRISMA guidelines. All necessary details are included and well discussed.
There are only some minor aspects:
- please introduce all abbreviations at first occurence
- typo issue in Table 2 (study 1 is printed in bold font)
- I would suggest sorting the abbreviations in the table legends alphabetically
- sometimes PT and CST is used, how do they differ?
Minor spell checking is required
